# Core outcome set for preventive intervention trials in chronic and episodic migraine (COSMIG): an international, consensus-derived and multistakeholder initiative

Kirstie Haywood ,[1] Rachel Potter ,[2] Robert Froud,[3] Gemma Pearce,[4] Barbara Box,[2] Lynne Muldoon,[2] Richard Lipton,[5] Stavros Petrou,[6] Regina Rendas-Baum,[7] Anne-Marie Logan,[8] Kimberley Stewart,[2] Martin Underwood ,[2] Manjit Matharu,[9] on behalf of the CHESS COSMIG group

For numbered affiliations see end of article.

**Correspondence to**
Dr Rachel Potter;
R.Potter@warwick.ac.uk

## ABSTRACT

**Objective** Typically, migraine prevention trials focus on reducing migraine days. This narrow focus may not capture all that is important to people with migraine. Inconsistency in outcome selection across trials limits the potential for data pooling and evidence synthesis. In response, we describe the development of core outcome set for migraine (COSMIG).

**Design** A two-stage approach sought to achieve international, multistakeholder consensus on both the core domain set and core measurement set. Following construction of a comprehensive list of outcomes, expert panellists (patients, healthcare professionals and researchers) completed a three-round electronic-Delphi study to support a reduction and prioritisation of core domains and outcomes. Participants in a consensus meeting finalised the core domains and methods of assessment. All stages were overseen by an international core team, including patient research partners.

**Results** There was a good representation of patients (episodic migraine (n=34) and chronic migraine (n=42)) and healthcare professionals (n=33) with high response and retention rates. The initial list of domains and outcomes was reduced from >50 to 7 core domains for consideration in the consensus meeting, during which a 2-domain core outcome set was agreed.

**Conclusion** International and multistakeholder consensus emerged to describe a two-domain core outcome set for reporting research on preventive interventions for chronic and episodic migraine: migraine-specific pain and migraine-specific quality of life. Intensity of migraine pain assessed with an 11-point Numerical Rating Scale and the frequency as the number of headache/migraine days over a specified time period. Migraine-specific quality of life assessed using the Migraine Functional Impact Questionnaire.

## BACKGROUND

International guidelines for the conduct of preventive studies for both episodic

### Strengths and limitations of this study

► The research process and validity of results are strengthened by the co-collaboration with patient research partners throughout all stages of the research.
► A bespoke grading system to support the prioritisation of outcome domains between stakeholder groups (expert panels) is described.
► International, multistakeholder participation of patients, researchers, and a range of health professionals in the on-line- Delphi survey.
► Expert panel representation in the Delphi survey was largely from Europe and North America.
► The majority of participants in the face-to-face consensus meeting were from the UK.

migraine (EM) and chronic migraine (CM) specify that the primary outcome should be focused on migraine days, or for CM on moderate to severe headache days.[1] Reviews of clinical trials of populations with CM and EM have identified substantial inconsistencies in outcomes reporting alongside often poorly defined outcomes.[2 3] An important impact of these inconsistencies is to limit the potential for robust meta-analyses.[4 5] For example, a 2015 meta-analysis of drugs for the prophylaxis of migraine by Jackson *et al*[6] did not include data from the largest and most robust trial of topiramate for CM (n=307) that found a mean difference of 1.7 migraine/migrainous days per 28 days after 12 weeks.[7] The reviewers meta-analysed the data from two much smaller (n=32 and n=50), low quality studies, and reported an effect size of 8.4 headache

days, the outcome specified for the meta-analyses after 12 weeks. Data that cannot be interpreted or used can result in unacceptable and unethical research waste. There is also potential for selective outcomes reporting and associated reporting bias if consistent outcomes are not pre-specified.[8 9]

Improved consistency, accountability and transparency in outcome reporting can be achieved by using a core outcome set (COS), a small, standardised group of outcomes that should be measured and reported, as a minimum, in all effectiveness trials for a specific health area.[10–12]

Current international guidelines for conduct of prevention studies in EM or CM have not developed outcome reporting recommendations in line with the current best practice.[1 13] Notably, patient input is markedly absent from these guidelines.

We describe here the development of a multiple-stakeholder, internationally endorsed, consensus-based COS applicable to preventative intervention trials and research studies in adults with episodic or chronic MIGraine (COSMIG).

## METHODS

Two key stages in COS development are described as follows (figure 1)[14]:

► Stage 1: defining the core domain set (CDS): what to measure, that is, the minimum number of health domains that should be assessed. A domain describes the concept or 'aspect of health or a health condition that needs to be measured to appropriately assess the effects of a health intervention'.[14]

► Stage 2: recommending the core measurement set (CMS): how to measure, that is, the minimum set of assessment methods that adequately correspond to the CDS.

We prospectively registered COSMIG with the Core Outcomes Measures in Effectiveness Trials initiative (http://www.comet-initiative.org/studies/details/953).

### Patient and public involvement

Following good practice guidance (https://www.invo.org.uk/posttyperesource/before-you-start-involving-people/[15]), we worked collaboratively with our patient research partners, who all had experience of CM or EM, throughout all stages of the research.

The COSMIG core group consisted of clinicians with expertise in headaches and migraine (MM, MU and Brendan Davies), including two international members (RL and Rigor Jensen), research scientists with expertise in clinical trials, Delphi technique, health measurement and qualitative research (MU, KH, RF, RP, SP, Vivien Nichols, Shilpa Patel and KS) and patient research partners (GP, BB and LM). Regular meetings were held between all group members to discuss the methodology for the Delphi

study and the subsequent consensus meeting. The group met specifically between each Delphi round, to discuss results, confirm feedback and format for subsequent rounds.

### Stage 1: core domain set
#### Stage 1.1: developing a comprehensive domain list
We first identified potential domains from systematic reviews[2 3] and qualitative research.[16] Domains were written in plain English as online questionnaires: one questionnaire contained domains for episodic headache, and one for chronic headache. Questionnaires were piloted with the core team and researchers naïve to the study (n=12).

#### Stage 1.2: international modified-Delphi process
Our primary goal for our Delphi study was to refine and prioritise domains. The Delphi process seeks to establish consensus between a panel of experts following a structured process of questionnaire completion and systematic feedback.[17 18] The panels are not intended to be representative of all headache specialists or people with migraine (as is the case when sampling from a definable population). We defined two expert panels external to the core research team: one comprised of expert patients with a target of up to 50 with CM and 50 with EM; and a second panel (also up to 50) comprised of healthcare professionals and researchers, who were representative of their professions and well placed to implement study findings.[19] Professionals included neurologists, nurse specialists, general practitioners, allied health professionals, researchers and measurement experts. We sought consensus between experts on the CDS.

#### Patients
We asked 13 national/international organisations to advertise the study on their social media platforms. Interested participants (≥18 years old) contacted the research team. We asked participants to self-diagnose/classify their migraines as EM or CM. Patient participants completed EM or CM questionnaires depending on their self-diagnosis.

#### Professionals
We invited national and international healthcare professionals (neurologists, General Practitioners (GPs), nurses, psychologists, pharmacists and allied health professionals) and researchers (triallists, reviewers, health economists and measurement experts) involved in headache research, identified through professional societies and from published research to participate. They were asked to complete both questionnaires.

The Delphi process had three sequential rounds with participants completing each prior round eligible to complete the next. The Delphi study administration and hosting of the online questionnaires was managed by Clinvivo.

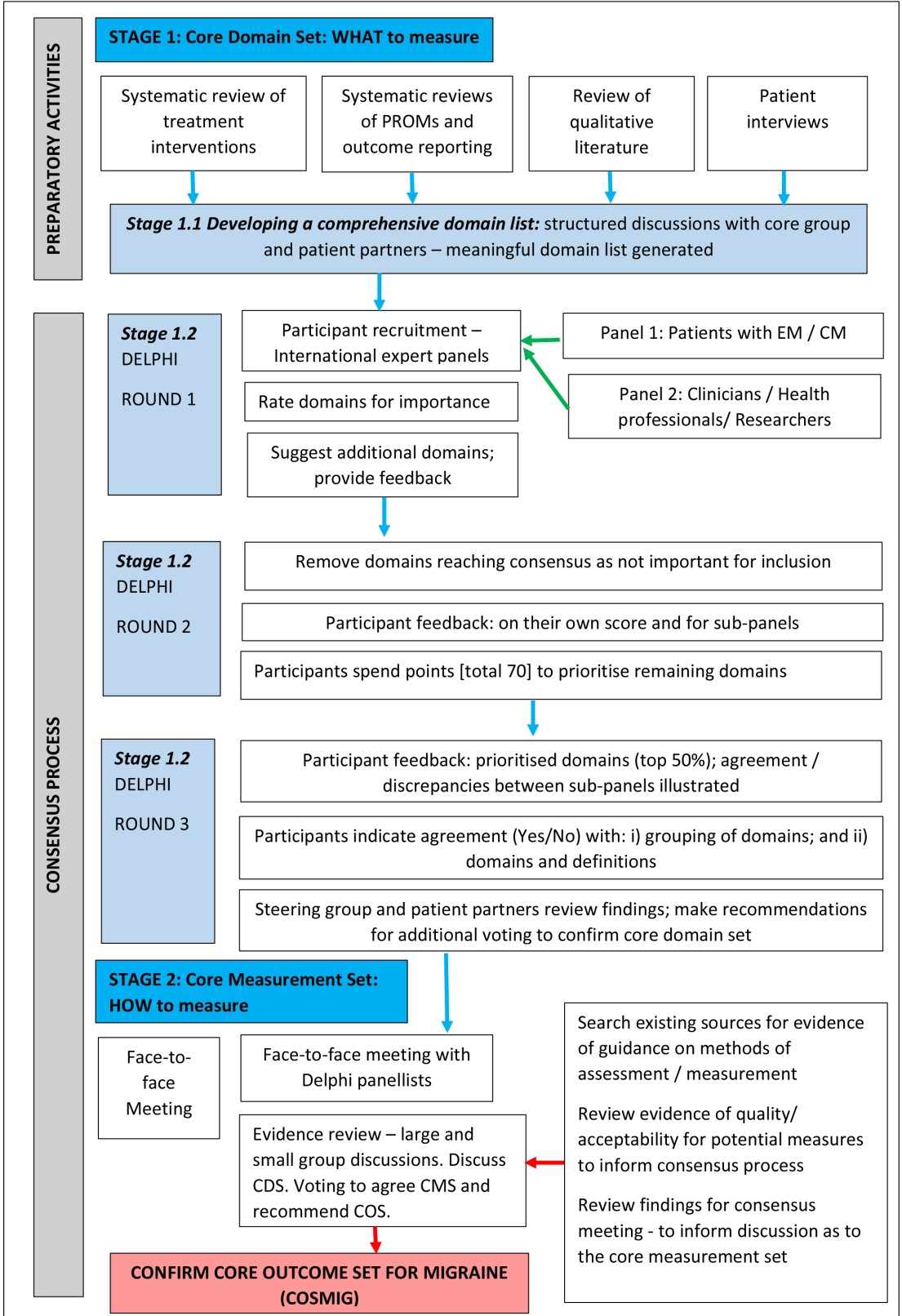

**Figure 1** Flow diagram outlining the development stages for the COSMIG. COSMIG, core outcome set with episodic or chronic MIGraine; CM, chronic migraine; EM, episodic migraine.

Round 1: participants rated the relative importance of each domain for inclusion in future research studies of chronic or episodic headache using a 9-point Numerical Rating Scale (NRS) (range: 1–3 'Not at all important', 4–6 'Uncertain' and 7–9 'Very important'). Participants could elaborate on their decisions by providing additional qualitative comment and/or provide additional domains for consideration

and rating in subsequent rounds. Informed by an approach described by Orbai *et al*,[20] we devised a bespoke grading system to illustrate where consensus was achieved and to indicate more easily where participants in each panel disagreed in their judgement (online supplemental table 1). An a priori decision rule determined that only those outcome domains judged most favourably by one or both panels (patients and professionals) would be included in round 2. That is, domains were included in round 2 if in both panels the median rating was 9 ('A\*\*'), or if in both panels ≥70% rated a domain ≥7 ('A\*'). If in both panels the median domain rating was ≥7 ('A'), or the median rating for a domain was ≥7 in just one panel ('B'), the domain could be included in round 2 if either panel achieved a median score of 9 or qualitative evidence supported further consideration.

Round 2: in round 2, we focused more specifically on migraine-specific domains (eg, nausea and photophobia), rather than headache-specific domains. Responses to round 1 were summarised and anonymous feedback provided. All participants received their own score for each domain, and the group median scores. Further prioritisation was achieved by inviting participants to 'spend points' (up to a maximum of 70) to illustrate how strongly they felt that a domain should be prioritised for inclusion in the CDS; a maximum of 10 points could be allocated to any 1 outcome domain (11-point scale: from 0 'Not a priority' to 10 'Absolute priority'). To ensure that subpanel differences were considered, and any discrepancies highlighted, the results from both panels were considered both separately and combined: the top 10 and top 50% of prioritised domains were discussed between COSMIG core team members, informing the maintenance of, or, where the concepts of health were similar, grouping of domains into a single 'meaningful' domain.

Round 3: responses to round 2 were summarised, highlighting the top 50% of prioritised domains and between-panel discrepancies. For those domains prioritised highly by just one panel (top 50%), participants were asked to reconsider if they should be included in the priority listing. If more than 70% of respondents selected 'yes', the domain was included. Finally, participants were asked to indicate by means of a dichotomous response if they: (a) were happy with the grouping of prioritised domains; (b) were happy with the proposed 'meaningful' domain and definition and (c) had additional comments. The frequency distribution of responses was calculated. Results from both subpanels were again considered separately and combined.

## Stage 2: core measurement set
### International expert panel face-to-face meeting
The purpose of the 1-day meeting was to discuss the CDS developed in our modified Delphi study, agree with the CMS and recommend the COS. Importantly,

participants were to consider that while a domain may be considered important, if an acceptable approach to measurement is not available, it is not appropriate to include the domain in a COS.

We invited professionals from Europe and patients from the UK who had taken part in our Delphi study. Participants received an information pack with meeting objectives and domain/measurement information ahead of the meeting. Where existing consensus for potential measures was not available, the COSMIG core team reviewed key data sources for guidance and evidence of measurement quality, acceptability and feasibility for use in preventive studies of EM or CM[21]:

► Migraine/headache:
  – Review of patient-reported outcome measures.[2]
  – International Headache Society guidelines.[1 13 22]
  – National Institute for Neurological Disorders Common Data Elements—Headache (preventative treatment).[23]
► Chronic pain and COS development:
  – Initiative on Methods, Measurement and Pain Assessment in Clinical Trials.[24–26]
  – Outcome Measures in Rheumatology and Clinical Trials group.[27]

The meeting started with an overview of the results of the Delphi process, prioritised domains and the evidence-based underpinning potential methods of assessment. Participants were asked to consider three options when determining domain 'placement' within the final COS[20]:

► Core 'inner' circle: domain is unambiguous with an acceptable method of assessment.
► Middle circle: domain is important, but not feasible for all preventative trials and research studies.
► Outer circle: domain is important but requires further study (research agenda)—for example, lacks conceptual clarity or method of assessment.

Semi-structured, small-group discussions with a mix of patients, healthcare professionals/researchers and members of the core research team (including patient partners) ensued, covering each prioritised domain. Two facilitators each supported two rounds of discussion per domain. Outcome domains and methods of assessment were reviewed in terms of importance, quality, acceptability and feasibility. Facilitators supported participant contribution and shared findings between groups to stimulate discussion. Following each small-group discussion, participants, with the exception of the core research team, were asked to indicate anonymously (paper-based questionnaire) their preference for domain inclusion (yes/no/do not know) and method of assessment (selecting one option from a short list) in the COS; an a priori definition of agreement required ≥70% of panellists to agree.

Next, small-group discussions and results were presented to the whole group. Where there was

**Table 1** Delphi round 1 shortlisted domains by voting prioritisation and agreement between panels

| Domain | Episodic migraine | | Chronic migraine | |
|---|---|---|---|---|
| | Evidence supporting inclusion in round 2 Delphi | | | |
| Section 1: life impact: symptoms associated with headache/migraine | Voting prioritisation | Qualitative feedback | Voting prioritisation | Qualitative feedback |
| Cognitive function: difficulty concentrating, ability to 'think clearly' or to remember things | (A) | Yes | A* | |
| Increased sensitivities: to light, sound, smell and touch | A* | | A* | |
| Pain associated with headache: experience an unpleasant physical sensation that aches or hurts | A** | | A* | |
| Duration of pain: associated with a headache | A** | | A* | |
| Frequency of pain: associated with a headache | A** | | A* | |
| Severity/intensity of pain: associated with a headache | A** | | A* | |
| Physical fatigue: experiencing physical fatigue, tiredness, lacking in energy and feeling physically exhausted | (A) | Yes | A* | |
| Sleep quality: being able to have a restful sleep | (A) | | A* | |
| Vomiting and/or feelings of nausea | A* | | (A) | |
| Anxiety: concerned, worried, fearful or anxious | (A) | Yes | (A) | Yes |
| Depressive mood: feeling sad, feeling down, feeling sorry for oneself or feeling depressed | (A) | Yes | (A) | No |
| Section 2: life impact: functioning, activities and general well-being | | | | |
| Activities of daily life | | | | |
| Being able to carry out usual tasks or daily activities inside or outside the home (not related to paid employment) that support an independent lifestyle—such as tidying one's home, walking short distances, managing finance, driving and using technology | (A) | | A* | |
| Needing to rest or lie down because of a headache | (A) | | A* | |
| Emotional well-being | | | | |
| Feelings of isolation: feeling isolated and reduced social interactions | (B) | Yes | (A) | Yes |
| Self-worth: feeling like a burden to others; can include feeling valued or helpless, accepted or rejected; and feelings of self-esteem | (B) | Yes | (A) | Yes |
| Stress: feelings of distress, frustration or irritation | A* | | (A) | Yes |
| Work/education | | | | |
| Being able to carry out activities related to work (paid or unpaid)/study to an acceptable or usual standard | A* | | A** | |
| Needing to take time-off work (paid or unpaid)/study | A* | | A* | |
| Social life | | | | |
| Social life: relationships with colleagues or peers | | | A* | |
| Family roles: being able to provide usual care and support for family and close friends | (A) | Yes | (A) | Yes |
| Participation in social or leisure activities: ability to participate in social or leisure activities | (A) | Yes | (A) | Yes |
| Overall health: an individual's general health status; the ability to live a 'normal' life | A* | | A* | |
| Self-management: ability to effectively decrease/minimise/control the impact of migraine on oneself (eg, pharmacology, diet and lifestyle choices) | A* | | A* | |

Continued

**Table 1** Continued

| Domain | Episodic migraine | | Chronic migraine | |
|---|---|---|---|---|
| | Evidence supporting inclusion in round 2 Delphi | | | |
| Unpredictability of a migraine: uncertainty of being symptom free or able to engage in activities | A* | No | (A) | Yes |
| Trigger factors: ability to avoid/manage migraine trigger factors | (B) | Yes | | No |
| Section 3: treatment effectiveness and financial impact | | | | |
| Satisfaction with treatment | A* | | A* | |
| Confidence in treatment | A* | | A* | |
| Consistency of treatment effect | A* | | A* | |
| Medication use: the type (potency) and dose (how much) medication taken when experiencing a migraine or headache | A* | | A* | |
| Medication use: the type (potency) and dose (how much) medication taken to prevent a migraine or headache | A* | | A* | |
| Financial impact: the economic cost associated with migraine treatment (to the individual (out of pocket expenses) and healthcare systems) | (A) | | A* | |
| Use of healthcare resources in response to migraine | (A) | | A* | |
| Section 4: complications (adverse events) | | | | |
| Treatment side effects: experiencing undesirable secondary effects from taking medications for migraine | A* | | A** | |
| Mortality (death) | (A) | | A** | |
| Included in round 2 due to importance scores (A** or A*) | 18 | | 24 | |
| Included in round 2 due to qualitative feedback | 9 | | 7 | |
| New outcomes added due to qualitative feedback | 0 | | 0 | |
| Total number of outcomes for inclusion in round 2 | 27 | | 31 | |

Each outcome was assigned to one of six categories reflecting levels of agreement: outcomes classified A** and A* would be included in round 2.
A*: if in both subpanels ≥70% rate an outcome is ≥7.
A**: if in both subpanels the median rating is 9.
(A): if in both subpanels the median outcome rating is ≥7.
(B): if the median rating for an outcome is ≥7 in only one subpanel.

agreement, no further discussion was required. Subsequent discussion focused on where further refinement was required. Finally, participants voted electronically to confirm domain placement in the COS (inner/middle/outer/out) and method of assessment. Proceedings were captured in the form of detailed written records and the outcomes of voting.

## RESULTS
### Stage 1: core domain set
#### Stage 1.1: developing a comprehensive domain list
A total of 57 (episodic) and 58 (chronic) domains were included in the questionnaire, grouped across four areas: symptoms (17), life impact (27 episodic/28 chronic), treatment effectiveness/financial impact (10) and complications (2). Piloting informed minor language modifications. Fifty-seven of the domains of interest were included for both EM and CM.

### Stage 1.2: international modified Delphi process
#### Round 1
Sub-panel 1 (patients): two organisations advertised the study (Migraine Association, Ireland, and National Migraine Centre, UK). Almost 80% (76/96) of patients who expressed an interest in taking part in the study completed the first questionnaire (42/53 CM (79%) and 34/43 EM (79%)). Most were female (CM: 40/53 (73%); EM: 29/43 (66%)) and aged between 36–45 years (CM: 41%) and 56–65 years (EM: 32%) (range: 18 –>66 years). Most were from the UK (57%), followed by the USA (19%), Ireland (14%), Canada (2%) and the rest of Europe (Denmark (2%) and France (5%)).

Sub-panel 2 (professionals): from a total of 198 international healthcare professionals/researchers invited to participate, 64 agreed. Nearly half (31/64 (48%)) joined the panel to complete the EM questionnaire; slightly more (33/64 (52%)) completed the CM questionnaires.

**Table 2** Delphi round 2. Results of domain prioritisation for episodic migraine (combined panels, n=27)†

| Rank* | Proposed 'merged' domain and definition | Top 10/27 prioritised domains | Top 50% of prioritised domains (rank 1–13/27 inclusive) | Lower 50% of prioritised domains (rank 14–27 inclusive) |
|---|---|---|---|---|
| 1 | Pain ► Experience of an unpleasant sensation that aches or hurts in the head; the frequency, severity and duration of this pain is important | Pain associated with migraine—experience of an unpleasant sensation that aches or hurts (1/27) Frequency of pain associated with a migraine (2/27) Severity or intensity of pain associated with a migraine (3/27) Duration of pain associated with a migraine (4/27) | | |
| 2 | Usual activities ► Being able to carry out usual activities (including paid or unpaid work, study, domestic chores, care or support for family or close friends) to an acceptable or usual standard ► Being able to participate in, or commit to, usual activities | Being able to carry out activities related to work (paid or unpaid) or study to an acceptable or usual standard (5/27) | Family roles—able to provide usual care or support for family or close friends, including ability to commit activities (11/27) Needing to take time off work (paid or unpaid) or study (13/27) | Participation in social or leisure activities—ability to participate in, or commit to, social or leisure activities (22/27) |
| 3 | Cognition ► Difficulty concentrating, ability to 'think clearly', or to remember things | Cognitive function—difficulty concentrating, ability to think 'clearly' or to remember things (6/27) | | |
| 4 | Adverse events | Treatment side effects—experiencing undesired secondary effects from taking medications for a migraine (7/27) | | |
| 5 | Overall health | An individual's general health status; the ability to 'live a normal life' (8/27) | | |
| 6 | Self-management | Trigger factors—the ability to avoid/manage migraine trigger factors (9/27) | Self-management—ability to effectively decrease/minimise/control the impact of migraine on oneself (eg, pharmaceutical, diet, lifestyle choices, etc) (11/27) | Unpredictability of a migraine—uncertainty of being symptom free or able to engage in activities (17/27) † prioritised in top 10 (10/27) by patients |
| 7 | Associated symptoms | Increased sensitivities—to light, sound, smell or touch (10/27) | | Vomiting and/or feelings of nausea (15/27)† prioritised in top 10 (8/27) by HCPs Physical fatigue—experiencing physical fatigue, tiredness, lacking in energy and feeling physically exhausted (18/27)† prioritised in top 50% (11/27) by patients |

Continued

**Table 2** Continued

| Rank* | Proposed 'merged' domain and definition | Top 10/27 prioritised domains | Top 50% of prioritised domains (rank 1–13/27 inclusive) | Lower 50% of prioritised domains (rank 14–27 inclusive) |
|---|---|---|---|---|
| 8 | Medication use | | | Satisfaction with treatment (14/27)† prioritised in top 10 (9/27) by HCPs |
| | | | | The type (potency) and dose (how much) of a medication taken when experiencing a migraine (16/27)† prioritised in top 50% (11/27) by HCPs |
| | | | | The type (potency) and dose (how much) of a medication taken to prevent a migraine (21/27) |
| | | | | Consistency in treatment (23/27) |
| | | | | Confidence in treatment (25/27) |
| 9 | Emotional well-being | | | Anxiety (19/27) |
| | | | | Depression (19/27)† prioritised in top 50% (13/27) by patients |
| | | | | Stress (24/27) |
| | | | | Self-worth (24/27) |
| | | | | Isolation (27/27) |

*Top 7 grouped domains: informed by top 10 and top 50% of prioritised domains (13/27).
†6 domains prioritised differently between the two panels, considered further in round 3.
HCPs, healthcare professionals.

Most were from the UK 14/33 (42%), with participants from the USA 5/33 (15%), Europe (Belgium 1/33 (3%), Germany 2/33 (6%), Italy 1/33 (3%), the Netherlands 1/33 (3%), Portugal 1/33 (3%), Serbia 1/33 (3%), Spain 2/33 (6%) and Turkey 1/33 (3%)), the Russian Federation 1/33 (3%), South Africa 1/33 (3%) and Thailand 1/33 (3%). Professionals included neurologists, nurse specialists, general practitioners, allied health professionals, researchers and measurement experts (online supplemental table 2).

In total, 75 (64%) and 65 (61%) panellists completed round 1 CM and EM questionnaires, respectively.

Most domains were rated as 'important', with few between panel discrepancies. Implementation of the a priori decision rule (online supplemental table 1) supported a 50% reduction in domains, with the prioritisation of 18/57 (episodic) and 24/58 (chronic) domains (table 1).

Qualitative feedback informed further consideration of 10 domains (9 episodic and 7 chronic) not achieving the proposed benchmark. No 'new' domains were proposed.

### Round 2
Round 2 questionnaires contained 27 episodic and 31 chronic domains (table 2). Round 2 was completed by 23/33 (70%) and 29/31 (93%) health professionals and 33/42 (79%) and 25/34 (74%) patients for CM and EM, respectively (totalling 54 EM (83%) and 56 CM (75%) questionnaires completed).

When prioritised according to the top 10 and top 50% of domains, several overriding 'meaningful' domains could be described (tables 2 and 3), six of which were common to both EM and CM: pain, usual activities, cognition, adverse events, overall health and associated symptoms. Respondents to the EM questionnaire also prioritised self-management, while medication use was prioritised by respondents to the CM.

Subpanel discrepancies for both EM and CM included patients' prioritisation of overall health, physical fatigue, unpredictability and self-management. Patients with EM also prioritised emotional well-being. Although awarded fewer points, people with CM prioritised the importance of social role and emotional well-being. In contrast, healthcare professionals prioritised treatment satisfaction, treatment side effects and vomiting/nausea for EM, and mortality and stress for CM.

### Round 3
Round 3 was completed by 23/23 (100%) and 21/29 (72%) health professionals, and 29/33 (88%) and 23/25 (92%) patients with CM and EM, respectively (totalling 52/56 (93%) for CM and 44/54 (81%) for EM). Discrepancies in six and three domains (top 10 or top 50% for one subpanel only) were considered for EM (treatment satisfaction, vomiting/feelings of nausea, medication taken during a migraine,

**Table 3** Delphi round 2. Results of domain prioritisation for chronic migraine (combined panels n=31)†

| Rank* | Domain and definition | Top 10/31 prioritised domains | Top 50% of prioritised domains (rank 1–15/31 inclusive) | Lower 50% of prioritised domains (rank 16–31 inclusive) |
|---|---|---|---|---|
| 1 | Pain<br>► Experience of an unpleasant sensation that aches or hurts in the head; the frequency, severity and duration of this pain is important | Severity or intensity of pain associated with a migraine (1/31)<br><br>Pain associated with a migraine—experience of an unpleasant sensation that aches or hurts (2/31)<br><br>Frequency of pain associated with a migraine (3/31)<br><br>Duration of pain associated with a migraine (4/31) | | |
| 2 | Usual activities<br>► Being able to carry out usual activities (including paid or unpaid work, study, domestic chores, care or support for family or close friends) to an acceptable or usual standard<br>► Being able to participate in, or commit to, usual activities | Being able to carry out usual tasks or daily activities inside or outside the home (not related to paid employment) that support an independent lifestyle—such as tidying one's home, walking short distances, managing finance, driving, usual technology (instrumental activities of daily life) (5/31)<br><br>Being able to carry out activities related to work (paid or unpaid) or study to an acceptable or usual standard (6/31) | Needing to take time off work (paid or unpaid) or study (11/31) | Family roles—able to provide usual care or support for family or close friends, including ability to commit activities (19/31)<br><br>Participation in social or leisure activities—ability to participate in, or commit to, social or leisure activities (22/31) |
| 3 | Cognition<br>► Difficulty concentrating, ability to 'think clearly', or to remember things | Cognitive function—difficulty concentrating, ability to think 'clearly' or to remember things (7/27) | | |
| 4 | Adverse events | Treatment side effects—experiencing undesired secondary effects from taking medications for migraine (8/31) | | Mortality (death) (26/31)† prioritised in top 50% (15/31) by HCPs |
| 5 | Associated symptoms | Increased sensitivities—to light, sound, smell or touch (9/31)<br><br>Physical fatigue—experiencing physical fatigue, tiredness, lacking in energy and feeling physically exhausted (10/31) | Sleep quality—being able to have a restful sleep (14/31)<br><br>Needing to rest or lie down because of a headache (15/31) | |
| 6 | Medication use | | Satisfaction with treatment (12/31) | The type (potency) and dose (how much) of a medication taken to prevent a migraine (21/31)<br><br>Consistency in treatment effect (23/31)<br><br>The type (potency) and dose (how much) of a medication taken during a migraine (24/31)<br><br>Confidence in treatment (28/31) |
| 7 | Overall health | | An individual's general health status; the ability to 'live a normal life' (13/31) | |

Continued

| Rank* | Domain and definition | Top 10/31 prioritised domains | Top 50% of prioritised domains (rank 1–15/31 inclusive) | Lower 50% of prioritised domains (rank 16–31 inclusive) |
|---|---|---|---|---|
| 8 | Emotional well-being | | | Stress—feelings of distress, frustration or irritation (16/31)† prioritised in top 10 (10/31) by HCPs |
| | | | | Anxiety—concerned, worried, fearful or anxious (20/31) |
| | | | | Self-worth—feeling like a burden to others; can include feeling valued or helpless; accepted or rejected; and feelings of self-esteem (28/31) |
| | | | | Feelings of isolation—feeling isolated and reduced social interactions (29/31) |
| | | | | Social role—relationship with work colleagues or peers (31/31) |
| 9 | Self-management | | | Self-management—ability to effectively decrease/minimise/control the impact of a migraine on oneself (eg, pharmaceutical, diet, lifestyle choices, etc) (17/31) |
| | | | | Unpredictability of a migraine—uncertainty of being symptom free or able to engage in activities (18/31)† prioritised in top 50% (14/31) by patients |
| 10 | Financial impact | | | Economic cost associated with treatment for a headache (to the individual (out-of-pocket expenses) and healthcare system) (25/31) |
| | | | | Use of healthcare resources in response to headache (30/31) |

*Top 5 grouped domains—informed by top 10 prioritised domains. Top 7 grouped domains—informed by top 13 and top 50% of prioritised domains (15/31).
†3 domains prioritised differently between the 2 panels; considered further in round 3.
HCPs, healthcare professionals.

unpredictability, physical fatigue and depressive mood) and CM (stress, mortality and unpredictability), respectively (online supplemental table 3).

The seven domains for EM were retained (>76% across sub-panels; >84% combined) (table 4) and a new domain 'treatment satisfaction' proposed (>70% healthcare professionals; 68% combined) (online supplemental table 3). Voting on subpanel discrepancies further supported the inclusion of vomiting/feelings of nausea, physical fatigue and depressive mood within the developing CDS for EM (online supplemental table 3). Qualitative feedback in the questionnaire supported a more positive rephrasing of the concept of self-management.

Six of the seven domains for CM were retained (>73% across subpanels; >80% combined) (table 4). 'Medication use' was rejected (<70%), and a redefining as 'treatment satisfaction' proposed. Qualitative feedback also highlighted the omission of 'visual disturbances' from 'associated symptoms', and the movement of 'sleep quality' to 'usual activities'.

For both EM and CM, qualitative feedback highlighted the importance of communication difficulties within cognitive function; further consideration of vomiting/nausea, fatigue and depressive mood as additional 'associated symptoms'; and unpredictability and ability to uphold usual commitments within 'usual activities'. Further clarification of the concept of 'overall health'—for example, general or migraine-specific health—was proposed and adoption of a standardised definition of 'adverse events' (Common Terminology Criteria for Adverse Events).[28]

The process defined seven core domains common to EM and CM (table 4). Additionally, EM included 'self-management'.

### Stage 2: core measurement set
#### International expert panel face-to-face meeting
The 1-day meeting took place at Warwick University in December 2018. Seven patients (three with EM and four with CM) and seven healthcare professionals/researchers (two doctors, two nurses, one physiotherapist and two measurement experts) participated from two countries (UK and Portugal). Ten core group members, including two patient research partners (GP and BB), attended.

#### *Pain*
Pain was redefined as migraine-specific pain and endorsed as an inner core domain for EM and CM (>70%) (table 5; figure 2). Based on review of existing measures and group discussion, voting supported recommendation of the 11-point NRS for assessing pain intensity[29] and number of headache/migraine

 Haywood K, *et al. BMJ Open* 2021;**11**:e043242. doi:10.1136/bmjopen-2020-043242

**Table 4** Delphi round 3. Results of voting for domains for EM and CM

| Proposed core domains for EM and CM (For voting in round 2) | Proposed 'meaningful domain' and definition (bold text informed by R3 qualitative feedback) | Q | EM Voting | | | CM Voting | | |
|---|---|---|---|---|---|---|---|---|
| | | | Patient (n=23) | HCPs (n=21) | Combined (n=44) | Patient (n=29) | HCPs (n=23) | Combined (n=52) |
| ▲ Pain associated with a migraine—an unpleasant sensation that aches or hurts ▲ Frequency of pain associated with a migraine ▲ Severity or intensity of pain associated with a migraine ▲ Duration of pain associated with a migraine | **Pain** ▲ Experience of an unpleasant sensation in the head that aches or hurts and is associated with experiencing a migraine ▲ The components of frequency, severity and duration of pain are all important Qualitative feedback supported the addition of: ▲ Unpleasant sensation in the head … **face, neck and/or shoulders …** | (a) (b) | 100.0% 82.6% | 100.0% 100.0% | 100.0% 90.9% | 96.6% 89.7% | 86.9% 95.7% | 92.3% 92.3% |
| ▲ An individual's health status; the ability to live a 'normal' life | **Overall health** ▲ An individual's health status; the ability to live a 'normal' life Qualitative feedback challenged the concept of 'normal life' and the lack of clarity with regards to either migraine-specific or general quality of life. This would be further explored during the consensus meeting. | (a) (b) | 100.0% 87.0% | 90.5% 81.0% | 95.5% 84.1% | 96.6% 89.7% | 87.0% 78.3% | 92.3% 84.6% |
| ▲ Being able to carry out activities related to work (paid or unpaid) or study to an acceptable or usual standard ▲ Family roles—able to provide usual care or support for family or close friends, including to commit to activities (EM only) ▲ Need to take time off work (paid or unpaid) or study ▲ Being able to carry out usual tasks or daily activities inside or outside the home (not related to employment) that support an independent lifestyle—such as tidying one's home, walking short distances, managing finance, driving and using technology (CM only) | **Usual activities** ▲ Being able to carry out usual activities (including paid or unpaid work, study, domestic chores, family or leisure activities, care or support for family or close friends) to an acceptable or usual standard ▲ Being able to participate in or commit to usual activities Qualitative feedback supported the importance of including 'unpredictability' in the definition: ▲ **Being able to plan, commit to, or participate in usual activities, including work, usual social or caring roles (due to the unpredictability of a migraine)** | (a) (b) | 95.7% 95.7% | 81.0% 76.2% | 88.6% 86.4% | 100.0% 89.7% | 95.7% 95.7% | 98.1% 92.3% |
| ▲ Cognitive function—difficulty concentrating, ability to think 'clearly' or to remember things | **Cognitive function** ▲ Difficulty with concentrating, thinking clearly or remembering things Qualitative feedback supported the addition of: ▲ **Difficulty with communication (word finding, slow or slurred speech)** | (a) (b) | 95.7% 91.3% | 100.0% 90.5% | 97.7% 90.9% | 96.6% 93.1% | 95.7% 95.7% | 96.1% 94.2% |

Continued

**Table 4** Continued

| Proposed core domains for EM and CM (For voting in round 3) | | EM Voting | | | CM Voting | | |
|---|---|---|---|---|---|---|---|
| ▲ Treatment side effects—experiencing undesired secondary effects from taking medications for migraine | | | | | | | |
| **Averse effects**<br>▲ Experiencing undesired secondary effects from taking medications for a migraine<br>Qualitative feedback supported adoption of the CTCAE standardised definition of adverse events:<br>▲ **'any unfavourable and unintended sign, symptom, or disease temporarily associated with the use of a medical treatment or procedure that may or may not be considered related to the medical treatment or procedure.' (CTCAE reference)** | (a)<br>(b) | 100.0%<br>87.0% | 100.0%<br>90.5% | 100.0%<br>88.6% | 89.7%<br>93.1% | 95.7%<br>82.6% | 92.3%<br>88.5% |
| ▲ Increased sensitivities—to light, sound, smell or touch<br>▲ Physical fatigue—experiencing physical fatigue, tiredness, lacking in energy and feeling physically exhausted (CM only)<br>▲ Sleep quality—being able to have a restful sleep (CM only)<br>▲ Needing to rest or lie down because of a headache (CM only) | | | | | | | |
| **Associated symptoms**<br>▲ Increased sensitivities—to light (**photophobia**), sound (**phonophobia**), smell, touch or movement<br>▲ Physical fatigue—experiencing physical fatigue, tiredness, lacking in energy and feeling physically exhausted (CM only)<br>▲ Sleep quality—being able to have a restful sleep (CM only)<br>▲ Needing to rest or lie down because of a headache (CM only)<br>Qualitative feedback highlighted concern over the omission of the following components from associated symptoms:<br>▲ Visual disturbances<br>▲ Depressive mood<br>▲ Vomiting/feelings of nausea<br>All to be explored in consensus meeting (for both EM and CM) | (a)<br>(b) | 87.0%<br>87.0% | 100.00%<br>90.5% | 93.2%<br>88.6% | 96.6%<br>93.1% | 73.9%<br>73.9% | 86.5%<br>84.6% |
| ▲ Satisfaction with treatment | | | | | | | |
| **Medication use**<br>**voting: proposed domain rejected (values <70%)**<br>Qualitative feedback highlighted the importance of a domain that was not just focused on medication use.<br>Note: voting on subgroup discrepancies supported the inclusion of 'treatment satisfaction' as a domain within the EM domain set.<br>Core group recommendation that 'treatment satisfaction' is explored in consensus meeting for both EM and CM | (a)<br>(b) | N/A<br>N/A | N/A<br>N/A | N/A<br>N/A | 79.3%<br>72.4% | 69.6%<br>60.9% | 75.0%<br>67.3% |

Continued

**Table 4** Continued

| Proposed core domains for EM and CM (For voting in round 3) | | EM Voting | CM Voting |
|---|---|---|---|
| ▲ Trigger factors—the ability to avoid/manage migraine trigger factors<br>▲ Self-management—the ability to effectively decrease/minimise/control the impact of a migraine on oneself (eg, by pharmaceutical, diet, lifestyle choices, etc) | **Self-management**<br>▲ Ability to effectively decrease/minimise/control the impact of a migraine on oneself (eg, by pharmaceutical, diet, lifestyle choices, etc)<br>▲ Ability to avoid/manage migraine trigger factors<br>Qualitative feedback—proposed a more positive definition:<br>▲ **Living better with migraine through lifestyle, dietary, pharmaceutical choices and taking an active part in long-term management of migraine with education and support**<br>▲ **Enabling patients to become active partners in their migraine treatment** | (a) 95.7%<br>(b) 91.3%<br><br>85.7%<br>81.0% | N/A<br>N/A<br><br>N/A<br>90.9%<br>86.4%<br>N/A<br>N/A |

Participants were invited to vote (Yes/No). (a) Are you happy with the grouping of prioritised domains (Yes/No)? (b) Are you happy with the proposed 'meaningful' domain and definition (Yes/No)?
Panellists did not vote in this domain.

CM, chronic migraine; CTCAE, Common Terminology Criteria for Adverse Events; EM, episodic migraine; HCP, Healthcare Professionals; N/A, Not applicable.

days per month for pain frequency.[1 22] Due to the complexities around the concepts of headache and migraine, it was recommended that the specific terminologies should be defined by individual studies.

### Overall health
Overall Health was redefined as 'migraine-specific quality of life' (MSQoL), endorsed as an inner core domain for both EM and CM (table 5; figure 2). Presented with evidence for generic and migraine quality of life measures, participants preferred the Migraine Functional Impact Questionnaire (MFIQ).[2 30] The four domain scores of the MFIQ address several key concepts highlighted throughout the COSMIG process—including usual activities, physical, cognitive, social and emotional functions. It also provides a global item score for usual activities.

### Pain duration and associated symptoms
Pain duration and associated symptoms were both judged as important but not feasible for inclusion in all trials/research studies and placed in the middle circle (table 5; figure 2).

### Self-management and treatment satisfaction
Self-management an treatment satisfaction were both considered important for both EM and CM, but lack of conceptualisation and assessment supported their placement on the research agenda (outer circle) (table 5; figure 2).

### Cognitive function and usual activities
Cognitive function and usual activities were both rejected as independent core domains but proposed as important components of MSQoL (table 5).

### Adverse events
Adverse event was rejected as a core domain, with the proposition that such reporting should be part of good clinical practice guidance (table 5; figure 2).

The result was a two-domain COSMIG (table 5; figure 2):
1. Migraine-specific pain: intensity assessed with the 11-point NRS and frequency as the number of headache/migraine days over a specified period.
2. MSQoL: assessed with the MFIQ.[30]

### DISCUSSION
The COSMIG process has identified two core domains—pain and MSQoL—that are recommended as part of a priori designated outcomes in future preventive intervention clinical trials for both EM and CM. Pain assessment should include both intensity measured with an 11-point NRS, and frequency assessed as the number of headache/migraine days per 28 days. SQoL should be assessed with the MFIQ.[30] Complex concepts around headache and migraine meant that participants in the consensus meeting were not able to make recommendations for the phrasing

**Table 5** Consensus meeting. Results from small and large group discussions and voting

| Domain | Small group | Large group | Final decision* |
|---|---|---|---|
| Pain | Domain<br>Voting supported inclusion of pain for EM and CM (>70%)<br>Three aspects of pain included:<br>▶ Intensity (11/11)<br>▶ Frequency (10/11)<br>▶ Duration (8/11)<br>Proposed domain refinement to 'migraine-specific pain'<br>Measurement:<br>Voting for individual options did not exceed 70%<br>Preferred assessments:<br>Intensity: 11-point NRS (55%)<br>Frequency: number of headache/migraine days (64%)<br>Duration: cumulative hours per 28 days of moderate/severe pain (55%) | Domain<br>Inner core: migraine-specific pain (no further voting required)<br>Measurement<br>Pain intensity: 11-point NRS (80%)<br>Pain frequency: the number of headache/migraine days (>70%)<br>Pain duration: no consensus. Proposed that daily capture (using paper or electronic diary) or retrospective capture using a questionnaire may not be feasible for all trials<br>Voting: middle circle (89%) | Domain—both EM and CM<br>Inner core: migraine-specific pain<br>Components: intensity and frequency<br>Measurement<br>Pain intensity: 11-point NRS (anchors 'no pain' and 'pain as bad as you can imagine')<br>Pain frequency:<br>the number of headache/migraine days<br>Pain duration: middle circle: important but not feasible for all trials/research studies |
| Overall Health | Domain<br>Voting supported redefining domain as migraine-specific quality of IMSQoLife (73%)<br>Measurement<br>MFIQ (72%) | Domain<br>Inner core: MSQoL (no further voting required)<br>Measurement<br>MFIQ | Domain—both EM and CM<br>Inner core: MSQoL<br>Measurement<br>MFIQ |
| Adverse events | Domain<br>Voting supported the rejection of adverse events from the core domain set (82%)<br>Measurement<br>N/A | Domain<br>Recommendations were supported. Should be captured as part of good clinical practice guidance | Not included in the COS for EM or CM |
| Self-management | Domain<br>No consensus on the inclusion (46%)/exclusion (54%) of self-management. Participants considered it to be important to both EM and CM, but requiring greater conceptualisation before it can be accurately measured | Domain<br>Group confirmed the importance of self-management for both EM and CM, but agreed that the lack of conceptualisation and method of assessment prevented inclusion in the COS<br>Voting: research agenda (73%) | Domain and measurement—both EM and CM<br>Outer circle: research agenda: important but requiring further study |
| Cognitive function | Domain<br>Voting supported the rejection of cognitive function as a separate core domain (70%)<br>but participants supported cognitive function as an important concept | Domain<br>Recommendations supported. The importance of cognitive function was supported and the potential for it to be captured with MSQoL proposed | Not included as a separate core domain for EM or CM.<br>Cognitive function is included within the new domain MSQoL' and will be assessed by the MFIQ |
| Associated symptoms | Domain<br>No consensus on the inclusion (50%)/exclusion (50%) of associated symptoms.<br>Participants discussed the importance of a wide range of associated symptoms—but capture of all would not be feasible in all trials (and hence not core) | Domain<br>Participants recognised pain as an important 'associated symptom' and the inclusion of several additional associated symptoms within the new domain 'MQoL' (captured by the MFIQ). Capturing a larger number of associated symptoms, or specific additional symptoms—such as fatigue—should be study specific and not core<br>Voting: middle circle (100%) | Domain and measurement—both EM and CM<br>Middle circle: important but not feasible to include in all trials/research studies |
| Usual activities | Domain<br>Voting supported the inclusion as a component of a new domain 'MQoL' (100%)<br>Measurement<br>Usual activities, as a component of MQoL to be assessed with the MFIQ (80%) | Domain<br>Recommendations were supported<br>Measurement<br>N/A | Not included as a separate core domain for EM or CM.<br>Usual activities is included within the new domain 'MSQoL' and will be assessed by the MFIQ |

Continued

**Table 5** Continued

| Domain | Small group | Large group | Final decision* |
|---|---|---|---|
| Treatment satisfaction | Domain<br>Considered important—but no consensus on the inclusion (64%)/exclusion (36%) of treatment satisfaction due to need for greater clarity | Domain<br>Group confirmed the importance of treatment satisfaction for both EM and CM, but agreed that the lack of conceptualisation and method of assessment prevented inclusion in the COS<br>Voting: research agenda (100%) | Domain and measurement—both EM and CM<br>Outer circle: research agenda: important but requiring further study |

*Core 'inner' circle: domain is unambiguous with an acceptable method of assessment. Middle circle: domain is important, but not feasible for all preventative trials and research studies. Outer circle: domain is important, but requires further study (research agenda)—for example, lacks conceptual clarity or method of assessment.
CM, chronic migraine; COS, core outcome set; EM, episodic migraine; MFIQ, Migraine Functional Impact Questionnaire; MSQoL, migraine-specific quality of life; NRS, Numerical Rating Scale.

of questions on pain severity (eg, worst, average or typical) or the definition of a migraine/headache day. Thus, the specific terminologies should be defined, and reported, by the needs of individual studies. Likewise, the specific timing of assessments should be driven by the requirements of the study.

Participants in the consensus meeting preferred the MFIQ over other measures of migraine-related quality of life such as the MSQoL Questionnaire V.2.1 and the 6-item Headache Impact Test (HIT-6) because participants, in particular, patient participants, felt its domains best reflected the impact migraine has on people's lives. This matches the aims of the original developers who specifically sought to address gaps in existing patient-reported outcomes.[31] A licence is needed to use the MFIQ available from Legal@evidera.com. The owners advise us that it will be available free of charge for non-commercial research (email, Evidera, 15 May 2020, personal communication). Pain duration and associated symptoms are important but are not considered core. How to assess self-management

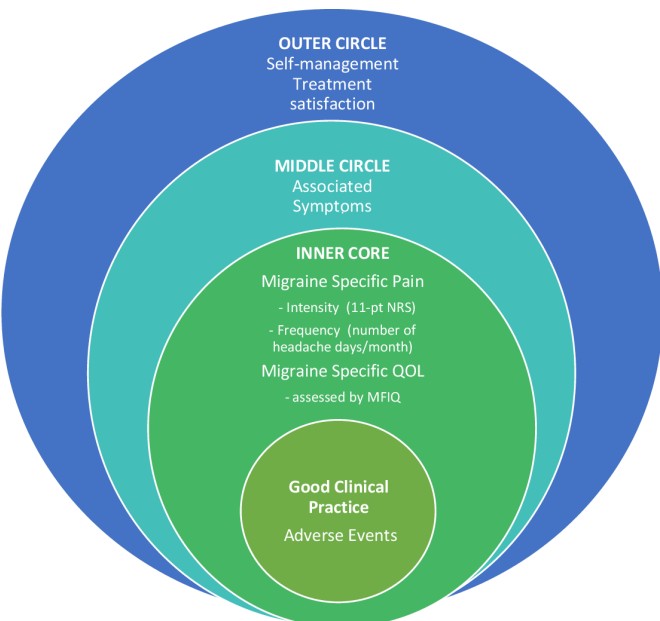

**Figure 2** COSMIG: core outcome set for episodic and chronic migraine.

and treatment satisfaction requires further research before recommendations can be made.

Our recommendation to include a reduction in the severity (intensity) and frequency in migraine pain is further supported by a recent modified Delphi study conducted in the USA, which sought to identify outcomes for value-based contracting for migraine medications.[32] However, a Delphi study of experts (N=12), published after our work was completed, focused on establishing the most useful outcome measures, specifically for non-pharmacological interventions for migraine, and identified the Migraine Disability Assessment followed by the HIT-6 as preferred outcomes.[33] Our empirical work does not support this prioritisation of outcome measures.[2 34]

The COSMIG recommendations contrast with previous guidance for trials of prophylaxis in CM that recommend a single primary outcome derived from headache/migraine days. Patient-reported headache-related quality of life appears last in order of the secondary outcomes[1] and guidelines for trials of prophylaxis in EM do not include quality of life as an outcome.[13] Informed by current good practice guidance in COS development,[9 14] this study included international participation from patient and professional panellists in an online Delphi study and a subsequent face-to-face meeting. All data pertaining to the Delphi study were analysed both separately and combined to ensure that the views of subpanels were clearly reported. This approach highlighted the value placed on patient-reported outcomes such as pain and quality of life by patients and health professionals. However, discrepancies pertaining to, for example, the importance of fatigue, unpredictability, emotional impact and cognitive function were described. Such discrepancies have been reported in other long-term musculoskeletal conditions[35] and more recently in a survey of health professionals and patients with COVID-19.[36] Evidence of such discrepancies is a key driver for the suggestion that patients' views are given at least equal wight to those of professionals in the process of COS development.[9] Incorporating outcomes that have resonance to all stakeholders can enhance trial relevance, providing valued information to inform decision-making in clinical practice and health policy settings.

 15

While individuals from 14 countries were included in the Delphi study, participants from just two countries (England and Portugal) contributed to the face-to-face meeting. However, both the Delphi process and consensus meeting sought input from credible 'experts'.[17] [19] For patients, expert is defined by experience of living with CM or EM, and for health professionals by their relative expertise in migraine-related research. The wide international involvement throughout the Delphi study improved international reach and helped ensure a wider relevance of the recommendations. We note that Delphi results are obtained from inviting experts to join a panel; as this eschews sampling, no inference should be made to any larger definable population.

Active pre-engagement with potential participants in the Delphi study enabled targeted follow-up of non-responders in round 1.[37] We note that the participation rate of invited panellists is higher than reported in some other Delphi studies, where response rates between 30% and 40% have been reported.[21] Moreover, a recent international Delphi study which sought to reach agreement on outcome measures for assessing the effectiveness of non-pharmacological interventions in migraine invited just 35 eligible researchers, as subject experts, and 4 patients.[33] Of the researchers, just 12 agreed to participate, with 10 (28%) completing all three rounds. This suggests that the focus of our Delphi study resonated with panellists, and, moreover, retention across subsequent rounds was high, with responses from both subpanels exceeding 70%.

More people with CM than with EM participated in the Delphi study, subpanel responses were analysed separately for both panels. Seven of the eight prioritised domains were common to both EM and CM; self-management was unique to EM. However, participants in the consensus meeting agreed that while poor conceptualisation and lack of assessment option prevented its consideration as a core domain, self-management was important for both EM and CM.

We relied on participant self-identification of diagnosis of EM/CM. Any misclassification is unlikely to have any substantive impact on our findings. The study included a broad age range of patient participants. Similarly, the healthcare professionals involved had a broad spectrum of experience in the care of patients with migraine and in migraine-related research.

Working collaboratively with patient research partners throughout the research contributed to the crafting of 'meaningful' domains at each stage of the Delphi process, giving validity to the proposed lists.[20] The initial Delphi questionnaire provided a comprehensive reflection of domains that might be assessed in CM or EM. Additional domains were not proposed by participants in round 1, supporting the comprehensiveness and relevance of content. Patient partners checked the comprehensibility and relevance of short-listed methods of assessment presented to participants in the consensus meeting, contributing to the debate and supporting lay participants during group discussions. All patient partners contributed to manuscript edits throughout the write-up phase.

The recommended COSMIG core set should be complemented by additional trial outcomes pertinent to the particular intervention being evaluated.[37] However, standardisation of core data collection is strongly advised to reduce the potential for systematic bias and enhance the quality of patient-reported outcomes data.[8] [9] More work is now needed on how to evaluate the self-management and treatment satisfaction domains.

Through an international collaboration between patients, researchers and health professionals, we have facilitated consensus on a COS for reporting on preventative intervention trials and research studies in adults with EM or CM (COSMIG). We recommend that both pain (intensity and frequency) and MSQoL are included as core domains. To support meaningful comparisons across studies, we recommend that pain intensity be assessed with an NRS[29] and frequency by determining the number of migraine days; MSQoL should be assessed with the MFIQ.[30] The timing of assessments should be determined by individual studies.

**Author affiliations**
[1]Warwick Research in Nursing, Division of Health Sciences, Warwick Medical School, University of Warwick, Coventry, UK
[2]Clinical Trials Unit, Warwick Medical School, University of Warwick, Coventry, UK
[3]Institute of Health Sciences, Kristiania University College, Oslo, Norway
[4]Department of Psychology and Behavioural Sciences, Coventry University, Coventry, UK
[5]Department of Neurology, Albert Einstein College of Medicine, Bronx, New York, USA
[6]Nuffield Department of Primary Care Health Sciences, Oxford, UK
[7]QualityMetric Incorporated, Lincoln, Rhode Island, USA
[8]Neurology Department, St George's University Hospitals NHS Foundation Trust, London, UK
[9]The Headache Group, National Hospital for Neurology and Neurosurgery, University College London Hospitals NHS Foundation Trust, London, UK

**Acknowledgements** We are very grateful to members of the International Steering Group (RL and Rigor Jensen) and the core COSMIG group (MU, MM, Brendan Davies, RP, SP, BB, GP, LM, RL, Rigor Jensen, Vivien Nichols, Shilpa Patel and Kimberley Stewart) for their oversight of the study. We are very grateful to everyone who participated in the Delphi study and attended the consensus meeting, and to Shilpa Patel and Vivien Nichols for their expert facilitation during the consensus meeting.

**Contributors** KH, MM, MU, RP, RF, RL, SP, BB, LM and GP made substantial contributions to the conception and design of the study and made substantial contributions to developing the protocol. KH, MM, MU, RP, RF, RL, RR-B, SP, AM-L, KS, BB, LM and GP made substantial contributions to the acquisition of data, analysis and interpretation of data. All authors have been involved in drafting the manuscript or revising it critically for important intellectual content, given final approval of the version to be published. KH will act as guarantor for the study.

**Funding** This study was funded by the National Institute for Health Research (NIHR) Programme Grants for Applied Research programme (RP-PG-1212-20018). The views expressed are those of the author(s) and not necessarily those of the NIHR or the Department of Health and Social Care. Active collaboration with our patient research partners was supported by a 'Delivering Results Award' (IAS/23022/16) from the Institute of Advanced Studies, Warwick University.

**Competing interests** MU and RF are directors and shareholders of Clinvivo and are part of an academic partnership with Serco related to return to work initiatives. MU recused himself from any discussions related to the choice of Delphi platform for this study; is a chief investigator or co-investigator on multiple previous and current research grants from the UK National Institute for Health Research and Arthritis Research UK, and is a co-investigator on grants funded by the Australian NHMRC; is a National Institute for Health Research (NIHR) senior investigator; has received travel expenses for speaking at conferences from the professional organisations hosting the conferences; is a co-investigator on two NIHR-funded

studies receiving additional support from Stryker; has accepted honoraria for teaching/lecturing from CARTA; and was an editor of the NIHR journal series and a member of the NIHR Journal Editors Group, for which he received a fee. MM serves on the advisory board for Abbott, Allergan, Eli Lilly, Medtronic, Novartis and TEVA; has received payment for the development of educational presentations from Allergan, electroCore, Eli Lilly, Medtronic, Novartis and TEVA; and has received research grants from Abbott, electroCore and Medtronic. SP is a director of Health Psychology Services, which, in part, provides psychological treatments for those with chronic pain.

**Patient consent for publication** Not required.

**Ethics approval** Ethical approval was gained from Warwick Medical School's Biomedical and Scientific Research Ethics Committee (approval number: REGO-2017–1921).

**Provenance and peer review** Not commissioned; externally peer reviewed.

**Data availability statement** Data are available upon reasonable request. De-identified data will be shard through the university accessible databases or repositories at Warwick University. Please contact KH if additional information is required: k.l.haywood@warwick.ac.uk.

**ORCID iDs**
Kirstie Haywood http://orcid.org/0000-0002-5405-187X
Rachel Potter http://orcid.org/0000-0001-6655-8996
Martin Underwood http://orcid.org/0000-0002-0309-1708

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
