## [Reviewer comments · BMJ Open]

ARTICLE DETAILS

TITLE (PROVISIONAL)	A Core Outcome Set for Preventive Intervention Trials in Chronic and Episodic Migraine (COSMIG): An international, consensus-derived and multi-stakeholder initiative.
AUTHORS	Haywood, Kirstie; Potter, Rachel; Froud, Robert; Pearce, Gemma; Box, Barbara; Muldoon, Lynne; Lipton, Richard; Petrou, Stavros; Rendas-Baum, Regina; Logan, Anne-Marie; Stewart, Kimberley; Underwood, Martin; Matharu, Manjit

VERSION 1 – REVIEW

REVIEWER	Simona Sacco University of L'Aquila, Italy
REVIEW RETURNED	04-Dec-2020

GENERAL COMMENTS	Authors performed a study to reach consensus on a core outcome set to improve reporting in migraine research. The Authors' work is potentially interesting and might have an important impact on future research in the migraine field. However, there are some limitations. 1) Less than half of invited healthcare professionals answered to the survey, as compared with about two thirds of the invited patients. Besides, patients outnumbered healthcare professionals in the present study. Those data suggest that healthcare professionals might not represent the overall cohort of professionals taking care of migraine patients. Besides, healthcare professionals were likely less motivated than patients to answer to the survey. Those points should be discussed. The results of the study should be interpreted with caution due to the low proportions of subjects answering to the survey. 2) Authors report that quality of life measures do not have adequate space in current guidelines for trials on migraine. In the present survey, patients were highly represented; therefore, a high importance given to patient-reported outcomes was expected. This point should be discussed. 3) Patients with chronic migraine outnumbered those with episodic migraine. Therefore, the Authors' suggestions mostly apply to those patients. This might be useful as patients with chronic migraine are the most difficult-to-treat; however, this point should be discussed.
--

REVIEWER	Christopher Gottschalk Yale Medicine USA
REVIEW RETURNED	31-Dec-2020

GENERAL COMMENTS	This is an extremely clearly written report concerning an important initiative to define a COS for migraine prophylaxis. The process was complex and time-consuming; digesting the steps and decisions also takes some effort. I think there is room for additional comment about the generalizability of the results, given that the vast majority of participants are from (what was until recently) the EU--only 14-15% of patients and professionals were from the US, despite the very large number of US migraine patients overall. An aspiration, though not a requirement, would be to have the authors provide more detail regarding the domains relegated to the "outer circle"--having reviewed so many aspects of migraine activity, burden, and experience over an extended period, I expect some specific aspects of these two domains, treatment satisfaction and self-management, could be discussed that would aid in their development and quantifiable measures.
---

REVIEWER	Robby De Pauw Ghent University, Belgium Sciensano, Belgium
REVIEW RETURNED	02-Mar-2021

GENERAL COMMENTS	I much appreciate the effort of the authors to find consensus for a core set of outcomes. As stated by the authors, the highest level of researcher, i.e. combining the results from different research groups to estimate a meta-analytic effect size has emerged as an important scientific indicator. Unfortunately, heterogeneity among studies often hinders such analysis. Common indicators are thus an important step to standardize research. Nevertheless, I have some concerns that I feel should be addressed before consideration of this study for publication. A first point of consideration is the COS. I think it is important to emphasize the importance of PROMs in this light? This is currently lacking in the manuscript. In addition, there is a more recent publication available on COS development: Prinsen, C.A., et al., How to select outcome measurement instruments for outcomes included in a "Core Outcome Set" - a practical guideline. Trials, 2016. 17(1): p. 449. Another important point is that the current researchers fail to address the most up-to date literature:  • Luedtke, K., Basener, A., Bedei, S., Castien, R., Chaibi, A., Falla, D., ... & Wollesen, B. (2020). Outcome measures for assessing the effectiveness of non-pharmacological interventions in frequent episodic or chronic migraine: a Delphi study. BMJ open, 10(2). • Swart, E. C., Good, C. B., Henderson, R., Manolis, C., Yanta, C., Parekh, N., & Neilson, L. M. (2020). Identifying Outcome Measures for Migraine Value-Based Contracting Using the Delphi Method. Headache: The Journal of Head and Face Pain, 60(10), 2139-2151. The methodology of the Delphi-study seems sound for most part. However, it has been recommended to have expert group/focus group panels beforehand to discuss the methodology of the Delphi-study (see Gibbs A. Focus groups. Soc Res Update 1997;19:1–8. and McMillan SS, King M, Tully MP. How to use the nominal group and Delphi techniques. Int J Clin Pharm 2016;38:655–62.) Has there been a focus group organized? Who was involved in this process? Can the stage 1.1 be considered such focus-group? What were the qualifications of the researchers involved?
--

	In stage 1.2 it is unclear to me if results of patients and professionals were combined or kept separate? The following statement is unclear for me: "An a priori decision rule determined that only those outcome domains judged most favourably by one or both groups (patients and professionals) would be included in round two." How was most favourably defined? Did you decide on an a priori cut-off? As I understand, participants already received specific domains. In a Delphi-study, the most common approach is to leave space in the first round for open answers to make sure the opinion of each participant is included. Was this the case? If not, why did the researchers deviate from this principle? By providing already specific domains, the participants are already directed towards a specific outcome. To me it is not clear what the difference is between migraine-specific and headache specific domains? Were these differences symptom-related (e.g. nausea, photophobia)? At stage 2, a panel discussion was organized, in which $\geq 70\%$ agreement was used as a cut-off. It is unclear to me if participants had the opportunity to review their original score or not, which is a common approach? The overview of results and discussion is well-constructed. However, a discussion of the results with results of other Delphi-studies is currently lacking.
--	--

VERSION 1 – AUTHOR RESPONSE

Reviewer: 1

Prof. Simona Sacco, University of L'Aquila Department of Clinical Sciences and Applied Biotechnology Comments to the Author:

Authors performed a study to reach consensus on a core outcome set to improve reporting in migraine research. The Authors' work is potentially interesting and might have an important impact on future research in the migraine field. However, there are some limitations.

Thank you for your positive comment.

Less than half of invited healthcare professionals answered to the survey, as compared with about two thirds of the invited patients. Besides, patients outnumbered healthcare professionals in the present study.

We acknowledge that it was disappointing that just 48% of those health professionals who originally agreed to take part in the Delphi study completed the first-round. However, a response rate of 48% is higher than that described in other studies, where response rates of between 30 and 40% have been reported [20. Prinsen et al, 2016]. Moreover, we note that a Delphi study is not the same as survey and that we are not making inferences from a sample to a population. For this reason, external validity cannot be damaged by response rate (as it can be in a survey sample) as external validity is not being claimed. Our approach is based on using an expert panel to make decisions rather than empirical evidence (such is the use of Delphi studies – i.e., when obtaining empirical evidence is either impractical or impossible) and we make no external generalisation from the panel to any larger definable population. We have now explained this point more clearly in our paper.

That said, regarding the panel response, panellists were asked to complete both CM and EM questionnaires, and it is conceivable that the length of both questionnaires was off-putting. However,

once engaged in the process, panel attrition was low. We have now noted this under limitations in the discussion (page 12):

We note that the participation rate of invited panellists is higher than reported in some other Delphi studies, where response rates between 30 and 40% have been reported.²⁰ Moreover, a recent international Delphi study which sought to reach agreement on outcome measures for assessing the effectiveness of non-pharmacological interventions in migraine invited just 35 eligible researchers as subject experts, and four patients.³² Of the researchers, just 12 agreed to participate, with 10 (28%) completing all three rounds. This suggests that the focus of our Delphi study resonated with panellists, and moreover, retention across subsequent rounds was high, with responses from both sub-panels exceeding 70%.

Regarding the point about the number of patients, we have added the following to the discussion:

Informed by current good practice guidance in core outcome set development^{9, 14}, this study included international participation from patient and professional groups in an on-line Delphi study and a subsequent face-to-face meeting. All data pertaining to the Delphi study were analysed both separately and combined to ensure that the views of sub-panels were clearly reported. This approach highlighted the value placed upon patient-reported outcomes such as pain and quality of life by patients and health professionals. However, discrepancies pertaining to, for example, the importance of fatigue, unpredictability, emotional impact, and cognitive function were described. Such discrepancies have been reported in other long-term musculoskeletal conditions³⁴ and more recently in a survey of health professionals and patients with COVID.³⁵ Evidence of such discrepancies is a key driver for the suggestion that patients' views are given at least equal weight to those of professionals in the process of core outcome set development.⁹ Incorporating outcomes that have resonance to all stakeholders can enhance trial relevance, providing valued information to inform decision-making in clinical practice and health policy settings.

Those data suggest that healthcare professionals might not represent the overall cohort of professionals taking care of migraine patients. Besides, healthcare professionals were likely less motivated than patients to answer to the survey. Those points should be discussed. The results of the study should be interpreted with caution due to the low proportions of subjects answering to the survey.

The Delphi approach does not seek a representative sample as is the objective in a survey but is rather seeking to establish the opinion of a panel of experts when empirical evidence is impossible or impractical to obtain. We have further sought to clarify this distinction in the manuscript.

We have added the following to the methods (page 4-5):

The Delphi process seeks to establish consensus between a panel of experts following a structured process of questionnaire completion and systematic feedback.^{17, 18} The panels are not intended to be representative of all headache specialists or people with migraine (as is the case when sampling from a definable population). We defined two expert panels external to the core research team: one comprised of expert patients with a target of up to 50 with chronic migraine (CM) and 50 with episodic migraine (EM); and a second panel (also up to 50) comprised of healthcare professionals and researchers, who were representative of their professions and well-placed to implement study findings [ADD FINK REF]. Professionals included neurologists, nurse specialists, general practitioners, allied health professionals, researchers, and measurement experts (Appendix Table 2). We sought consensus between experts on the core domain set.

And the following to the discussion:

Whilst individuals from 14 countries were included in the Delphi study, participants from just two countries (England and Portugal) contributed to the face-to-face meeting. However, both the Delphi process and consensus meeting sought input from credible 'experts'.^{17, 36} For patients, expert is defined by experience of living with chronic or episodic migraine, and for health professionals by their relative expertise in migraine-related research. The wide international involvement throughout the Delphi study improved international reach and helped ensure a wider relevance of the recommendations. We note that Delphi results are obtained from inviting experts to join a panel; as this eschews sampling, no inference should be made to any larger definable population.

2) Authors report that quality of life measures do not have adequate space in current guidelines for trials on migraine. In the present survey, patients were highly represented; therefore, a high importance given to patient-reported outcomes was expected. This point should be discussed. Thank you for this suggestion.

Our view is that the strength of the patient input into this study is one of its key strengths

As per good practice for COS development, we sought to engage with both patients and health professionals to explore the outcomes that mattered to both groups, and to subsequently achieve a consensus between both groups. Whilst the number of patients participating in the Delphi study did slightly outnumber those of health professionals, the results were analysed separately and combined throughout the Delphi process. This approach evidenced the fact that patient-reported outcomes were highly valued by both patients and health professionals. This was not a finding that was unique to patients.

We have added the following to the discussion:

All data pertaining to the Delphi study were analysed both separately and combined to ensure that the views of sub-panels were clearly reported. This approach highlighted the value placed upon patient-reported outcomes such as pain and quality of life by patients and health professionals. However, discrepancies pertaining to, for example, the importance of fatigue, unpredictability, emotional impact, and cognitive function were described. Such discrepancies have been reported in other long-term musculoskeletal conditions³⁴ and more recently in a survey of health professionals and patients with COVID.³⁵ Evidence of such discrepancies is a key driver for the suggestion that patients' views are given at least equal weight to those of professionals in the process of core outcome set development.⁹ Incorporating outcomes that have resonance to all stakeholders can enhance trial relevance, providing valued information to inform decision-making in clinical practice and health policy settings.

3) Patients with chronic migraine outnumbered those with episodic migraine. Therefore, the Authors' suggestions mostly apply to those patients. This might be useful as patients with chronic migraine are the most difficult-to-treat; however, this point should be discussed.

It is important to note that Delphi studies differ from survey work, in that the Delphi approach is intended to help establish expert consensus rather than measure people's views and make inferences about population parameters (as is the objective in a survey). However, to explore differences in views that may have warranted panel consideration, throughout the process, the panel responses from CM and EM patient experts were analysed separately. This highlighted where there was overlap and where discrepancies occurred.

We have added the following to the discussion:

More people with chronic migraine than with episodic migraine participated in the Delphi study, sub-panel responses were analysed separately for both panels. Seven of the eight prioritised domains were common to both episodic and chronic migraine; self-management was unique to episodic

migraine. However, participants in the consensus meeting agreed that whilst poor conceptualisation and lack of assessment option prevented its consideration as a core domain, self-management was important for both episodic and chronic migraine.

Reviewer: 2

Dr. Christopher Gottschalk, Yale School of Medicine Comments to the Author:

This is an extremely clearly written report concerning an important initiative to define a COS for migraine prophylaxis. The process was complex and time-consuming; digesting the steps and decisions also takes some effort.

Thank you for your kind and considered comments. These are much appreciated.

I think there is room for additional comment about the generalizability of the results, given that the vast majority of participants are from (what was until recently) the EU--only 14-15% of patients and professionals were from the US, despite the very large number of US migraine patients overall.

Many thanks for this comment.

Whilst it would have been preferable to have a greater number of experts from the US, we are not aware of any evidence that suggests that the experience of migraine differs for people living in the US, when compared to the EU. A member of our international steering group is an international recognised clinical academic from the US. The initial extensive list of outcomes was informed by reviews of the international literature (both quantitative and qualitative). No participant added additional outcomes at any stage of the Delphi process. Moreover, the Delphi method is not concerned with having a generalisable or representative sample. Rather, it seeks to facilitate consensus within an expert panel with specific expertise on a topic).

We have added the following to the discussion:

Whilst individuals from 14 countries were included in the Delphi study, participants from just two countries (England and Portugal) contributed to the face-to-face meeting. However, both the Delphi process and consensus meeting sought input from credible 'experts'.^{17, 36} For patients, expert is defined by experience of living with chronic or episodic migraine, and for health professionals by their relative expertise in migraine-related research. The wide international involvement throughout the Delphi study improved international reach and helped ensure a wider relevance of the recommendations. We note that Delphi results are obtained from inviting experts to join a panel; as this eschews sampling, no inference should be made to any larger definable population.

An aspiration, though not a requirement, would be to have the authors provide more detail regarding the domains relegated to the "outer circle"--having reviewed so many aspects of migraine activity, burden, and experience over an extended period, I expect some specific aspects of these two domains, treatment satisfaction and self-management, could be discussed that would aid in their development and quantifiable measures.

We thank the reviewer for their interest in these additional domains. However, we believe that additional comment on these domains is outside the scope of this particular manuscript and (as proposed by the reviewer) will not seek to address them any further in this paper. As indicated, this elaboration should be the focus of future work.

Reviewer: 3

Dr. Robby De Pauw, Ghent University. Comments to the Author:

I much appreciate the effort of the authors to find consensus for a core set of outcomes. As stated by the authors, the highest level of researcher, i.e. combining the results from different research groups to estimate a meta-analytic effect size has emerged as an important scientific indicator. Unfortunately, heterogeneity among studies often hinders such analysis. Common indicators are thus an important step to standardize research.

Nevertheless, I have some concerns that I feel should be addressed before consideration of this study for publication.

Thank you for your positive and insightful comments.

A first point of consideration is the COS. I think it is important to emphasize the importance of PROMs in this light? This is currently lacking in the manuscript. In addition, there is a more recent publication available on COS development:

Prinsen, C.A., et al., How to select outcome measurement instruments for outcomes included in a "Core Outcome Set" - a practical guideline. *Trials*, 2016. 17(1): p. 449.

The paper details the two core stages in COS development: stage 1, defining a core domain set (what to measure) and stage 2, defining a core measurement set (how to measure). The importance of PROMs is specific to this second stage.

Whilst completing this work we also completed a systematic review of the quality and acceptability of PROMs in the headache / migraine population. This is referred to in Stage 2 – Core Measurement Set. During the completion of this work we made close reference to the work of COSMIN, which is an integral component of the process referred to in the Prinsen paper (and indeed the lead author of this submission participated in the Prinsen study).

The shortlisting of methods of assessment was informed by the key issues addressed in the Prinsen paper – and we have now included a reference to this paper.

The COS has resulted in the recommendation of two PROMs – one for pain and the second for migraine-specific quality of life. Word count limits the extent to which we can expand on the discussion around PROMs, but we have now added the following to the discussion:

Our recommendation to include a reduction in the severity (intensity) and frequency in migraine pain is further supported by a recent modified-Delphi study conducted in the US, which sought to identify outcomes for value-based contracting for migraine medications.³¹ However, a Delphi study of experts (N=12) published after our work was completed focussed on establishing the most useful outcome measures, specifically for non-pharmacological interventions for migraine, identified the Migraine Disability Assessment (MIDAS) followed by the HIT-6 as preferred outcomes.³² Our empirical work does not support this prioritisation of outcome measures, 2,33

Another important point is that the current researchers fail to address the most up-to date literature:

- Luedtke, K., Basener, A., Bedei, S., Castien, R., Chaibi, A., Falla, D., ... & Wollesen, B. (2020). Outcome measures for assessing the effectiveness of non-pharmacological interventions in frequent episodic or chronic migraine: a Delphi study. *BMJ open*, 10(2).

- Swart, E. C., Good, C. B., Henderson, R., Manolis, C., Yanta, C., Parekh, N., & Neilson, L. M. (2020). Identifying Outcome Measures for Migraine Value-Based Contracting Using the Delphi Method. *Headache: The Journal of Head and Face Pain*, 60(10), 2139-2151.

Thank you for highlighting these two recent papers.

They have contributed to our discussion in the following way:

Page 13:

We note that the participation rate of invited panellists is higher than reported in some other Delphi studies, where response rates between 30 and 40% have been reported.²⁰ Moreover, a recent international Delphi study which sought to reach agreement on outcome measures for assessing the effectiveness of non-pharmacological interventions in migraine invited just 35 eligible researchers as subject experts, and four patients.³² Of the researchers, just 12 agreed to participate, with 10 (28%) completing all three rounds. This suggests that the focus of our Delphi study resonated with

panellists, and moreover, retention across subsequent rounds was high, with responses from both sub-panels exceeding 70%.

Page 12:

Our recommendation to include a reduction in the severity (intensity) and frequency in migraine pain is further supported by a recent modified-Delphi study conducted in the US, which sought to identify outcomes for value-based contracting for migraine medications.³¹ However, a Delphi study of experts (N=12) published after our work was completed focussed on establishing the most useful outcome measures, specifically for non-pharmacological interventions for migraine, identified the Migraine Disability Assessment (MIDAS) followed by the HIT-6 as preferred outcomes.³² Our empirical work does not support this prioritisation of outcome measures, 2,33

The methodology of the Delphi-study seems sound for most part. However, it has been recommended to have expert group/focus group panels beforehand to discuss the methodology of the Delphi-study (see Gibbs A. Focus groups. Soc Res Update 1997;19:1–8. and McMillan SS, King M, Tully MP. How to use the nominal group and Delphi techniques. Int J Clin Pharm 2016;38:655–62.) Has there been a focus group organized? Who was involved in this process? Can the stage 1.1 be considered such focus-group? What were the qualifications of the researchers involved?

Many thanks for highlighting this methodological approach.

We established both a core research team which included methodologists (with expertise in COS development, Delphi technique, and measurement science), clinicians (headache/migraine research), and patient research partners. This group was further supported by an international steering group of expert clinicians in headache / migraine research. This group met regularly to inform and discuss development of the Delphi study and discussed the results at each stage.

The role of the team has been enhanced in the methods section (page 4):

Patient and public involvement

Following good practice guidance [<https://www.invo.org.uk/posttyperesource/before-you-start-involving-people/>; 15, we worked collaboratively with our patient research partners, who all had experience of chronic or episodic migraine, throughout all stages of the research.

The COSMIG core group consisted of clinicians with expertise in headaches and migraine (MM,MU, BD), including two international members (RL,RJ), research scientists with expertise in clinical trials, Delphi technique, health measurement and qualitative research (MU,KH,RF,RP,SP,VN,SP,KS) and patient research partners (GP,BB,LM). Regular meetings were held between all group members to discuss the methodology for the Delphi study and consensus meeting. The group met specifically between each Delphi round, to discuss results, confirm feedback and format for subsequent rounds.

In stage 1.2 it is unclear to me if results of patients and professionals were combined or kept separate?

Thank you for seeking clarification on this important point.

We sought to ensure that panel differences were observed throughout the process detailed in stage 1.2.

In round 1 the bespoke grading system specifically considered the median scores per domain for each group. In round 2, the results from both groups were considered both separately and combined, thus highlighting important between group discrepancies which were considered further in round 3. In

round 3, the results across sub-panels and combined were considered. This highlighted if consensus was achieved both across sub-panels and across the combined population. We note that this was not clear in the methods section of round 3 (and only reported in the results section).

We have added more detail to the text with regards to the bespoke grading system used in round 1 (detailed in Appendix Table 1).

An a priori decision rule determined that only those outcome domains judged most favourably by one or both panels (patients and professionals) would be included in round two. That is, domains were included in round 2 if in both panels the median rating was 9 ('A**'), or if in both panels $\geq 70\%$ rated a domain ≥ 7 ('A*'). If in both panels the median domain rating was ≥ 7 ('A'), or the median rating for a domain was ≥ 7 in just one panel ('B'), the domain could be included in round 2 if either panel achieved a median score of 9 or qualitative evidence supported further consideration.

We have edited the following sentence in round 2:

To ensure that sub-panel differences were considered, and any discrepancies highlighted, the results from both panels were considered both separately and combined:

We have added the following sentence to round 3:

Results from sub-panels were again considered separately and combined.

The following statement is unclear for me: "An a priori decision rule determined that only those outcome domains judged most favourably by one or both groups (patients and professionals) would be included in round two." How was most favourably defined? Did you decide on an a priori cut-off? Thank you for requesting clarification on this point.

We have now added additional detail into the methods paragraph:

Informed by an approach described by Orbai et al. (2017) 17, we devised a bespoke grading system to illustrate where consensus was achieved and to indicate more easily where participants in each group disagreed in their judgement (Appendix Table 1). An a priori decision rule determined that only those outcome domains judged most favourably by one or both groups (patients and professionals) would be included in round two. That is, domains were included in round 2 if in both panels the median rating was 9 ('A**'), or if in both panels $\geq 70\%$ rated a domain ≥ 7 ('A*'). If in both panels the median domain rating was ≥ 7 ('A'), or the median rating for a domain was ≥ 7 in just one panel ('B'), the domain could be included in round 2 if either panel achieved a median score of 9 or qualitative evidence supported further consideration.

As I understand, participants already received specific domains. In a Delphi-study, the most common approach is to leave space in the first round for open answers to make sure the opinion of each participant is included. Was this the case? If not, why did the researchers deviate from this principle? By providing already specific domains, the participants are already directed towards a specific outcome.

However, participants were afforded the opportunity to add additional outcome domains to the list that was informed by the search and/or to provide qualitative feedback at every stage of the Delphi process. Whilst no additional outcome domains were proposed at any stage (which we believe supported the content validity of the list), qualitative feedback did help to develop the way in which domains were defined.

We have extended the following description in the methods section:

Participants could elaborate on their decisions by providing additional qualitative comment and/or provide additional domains for consideration and rating in subsequent rounds.

To me it is not clear what the difference is between migraine-specific and headache specific domains? Were these differences symptom-related (e.g. nausea, photophobia)?

The reviewer is correct that we focussed more on migraine symptoms such as nausea and photophobia. We have clarified this in the text.

Round 2 In round two we focused more specifically on migraine-specific (e.g. nausea and photophobia), rather than headache-specific, domains.

At stage 2, a panel discussion was organized, in which $\geq 70\%$ agreement was used as a cut-off. It is unclear to me if participants had the opportunity to review their original score or not, which is a common approach?

Many thanks for seeking clarification on this point.

As indicated in the methods section, responses to round 1 were summarised and anonymous feedback provided. Participants received both their own score and the group median score for each domain. We have edited this to make it clearer:

Responses to round one were summarised and anonymous feedback provided. Participants all received their own score for each domain, and the group median scores.

The overview of results and discussion is well-constructed. However, a discussion of the results with results of other Delphi-studies is currently lacking.

Many thanks for this recommendation. Within the available space allowed, we have now increased reference to other Delphi studies throughout the discussion.

VERSION 2 – REVIEW

REVIEWER	Sacco, Simona University of L'Aquila Department of Clinical Sciences and Applied Biotechnology, Neuroscience Section
REVIEW RETURNED	06-May-2021

GENERAL COMMENTS	I appreciated the responses and I don't have further comments.
--

REVIEWER	De Pauw, Robby Ghent University
REVIEW RETURNED	31-May-2021

GENERAL COMMENTS	The authors did an amazing job in amending the manuscript based on the reviewers' comments.
---